# A STATISTICAL APPROACH FOR CONTROLLED TRAINING DATA DETECTION

**Zirui Hu[1], Yingjie Wang[1]\*, Zheng Zhang[1], Hong Chen[2], Dacheng Tao[1]\***
[1]Generative AI Lab, College of Computing and Data Science, Nanyang Technological University
[2]College of Informatics, Huazhong Agricultural University
`zirui.hu@ntu.edu.sg, zhengzhang@mail.ustc.edu.cn`
`{yingjiewang1201, dacheng.tao}@gmail.com`
`chenh@mail.hzau.edu.cn`

## ABSTRACT

Detecting training data for large language models (LLMs) is receiving growing attention, especially in high-reliability applications. While numerous efforts have been made to address this issue, they typically focus on accuracy without ensuring controllable results. To fill this gap, we propose **K**nockoff Inference-based **T**raining data **D**etector (KTD), a novel method that achieves rigorous false discovery rate (FDR) control in training data detection. Specifically, KTD generates synthetic knockoff samples that seamlessly replace original data points without compromising contextual integrity. A novel knockoff statistic, which incorporates multiple knockoff draws, is then calculated to ensure FDR control while maintaining high power. Our theoretical analysis demonstrates KTD's asymptotic optimality in terms of FDR control and power. Empirical experiments on real-world datasets, such as WikiMIA, XSum, and Real-Time BBC News, further validate KTD's superior performance compared to existing methods.

## 1 INTRODUCTION

Large language models (LLMs) have achieved remarkable performance across a wide range of natural language processing (NLP) tasks, including machine translation (Wong et al., 2023), code completion (Chen et al., 2021; Li et al., 2022), and question answering (Dong et al., 2024; Li et al., 2024a). This success is largely driven by the use of massive language corpora, often reaching the trillion-token scale (Computer, 2023). While such extensive datasets equip LLMs with broad knowledge and strong text generation capabilities, they may also contain private information (Carlini et al., 2021) or copyrighted content (Chang et al., 2023) collected from the Internet, leading to potential risks. For instance, LLMs can inadvertently memorize and reproduce sensitive information when prompted with carefully crafted inputs, posing significant threats to privacy and intellectual property rights (Carlini et al., 2021).

To address this issue, recent studies have explored methods for detecting training data within LLMs (Shi et al., 2023; Golchin & Surdeanu, 2023). However, these approaches typically frame the problem as a binary classification task, focusing solely on distinguishing between training and non-training samples with high accuracy. We argue that accuracy alone is insufficient, particularly in cases involving copyright violation detection. For instance, if a copyright holder seeks legal action against a technology company for unauthorized use of proprietary content (Grynbaum & Mac, 2023), it is critical to ensure that the majority of identified training samples were genuinely used by the company for training. False detections in such cases could lead to unwarranted legal consequences. Therefore, beyond accuracy, controlling the false discovery rate (FDR) (Benjamini & Hochberg, 1995)—also referred to as the false positive rate in binary classification—is essential for training data detection and should not be overlooked.

In this paper, we study the problem of detecting training samples from LLMs with controllable FDR. Specifically, given a set of text samples, our goal is to determine whether the model has been trained on them while ensuring that the proportion of falsely identified training samples (i.e.,

---

*Corresponding authors

non-training samples mistakenly classified as training samples) remains within a predefined bound. Inspired by controllable variable selection, we propose a Knockoff Inference-based Training Data Detector (KTD), which treats the problem of detecting training samples as an instance of relevant variable selection, thereby enabling the use of knockoff inference's (KI) robust capacity in FDR controlling. KI generates knockoff variables corresponding to each original variable and computes knockoff statistics for each pair of original and knockoff variables, which are then used to identify relevant variables. Building on the KI paradigm, KTD operates in two stages. In the first stage, KTD generates knockoffs that preserve the semantics of the original text samples, ensuring they can substitute for the originals without altering the context. In the second stage, KTD computes knockoff statistics to capture differences between the original text samples and their corresponding knockoffs, using these differences as indicators to identify training samples.

In KTD, a fundamental component for successful detection is the construction of the knockoff statistic. While the knockoff statistic from the vanilla KI method can be directly applied to KTD, it has a significant drawback: the vanilla KI method uses only a single draw of the knockoff variable to calculate the knockoff statistic $W_j$. This approach results in a high variance in $W_j$ due to the inherent randomness of the knockoff process, making it difficult to distinguish between training and non-training samples. Such indistinguishability can lead to overly conservative detection results, excluding too many true training samples to achieve the desired FDR control. As a result, the detection efficiency—measured by power (i.e., the proportion of actual training samples correctly identified)—is significantly compromised.

To address this issue, KTD adopts a novel approach for calculating the knockoff statistic which utilizes multiple draws of the knockoff variable. This design effectively reduces the variance of the knockoff statistic, making it more centralized and thereby enhancing the separability between training and non-training samples. Specifically, KTD draws $m$ realizations of the knockoff variable and computes the knockoff statistic $\tilde{W}_j$ by comparing the importance score of the original variable to the average of the importance scores of these realizations.

To theoretically justify KTD, we first demonstrate that the knockoff statistic of KTD, i.e., $W_j^{\text{KTD}}$ retains the symmetric property, ensuring that it can control the FDR just like the vanilla KI. Furthermore, we distinguish KTD from the vanilla KI by proving its asymptotically optimal property: as $m$ approaches infinity, the power converges to 1 and the FDR converges to 0.

We empirically evaluate KTD using three popular large language models on three real-world datasets, including the established benchmark WikiMIA. The experimental results demonstrate that KTD not only achieves the desired FDR control without relying on a validation set, which existing methods depend on but also exhibits significantly higher power compared to vanilla KI when achieving similar FDR levels.

To summarize, our contributions are as follows:

1. We address the problem of detecting training data from LLMs with FDR control through the perspective of knockoff inference and propose a knockoff inference-based detecting method KTD.

2. We theoretically justify our proposed KTD from two aspects. Firstly, we prove that the knockoff statistic in KTD possesses the symmetric property, which is essential for effective FDR control. Secondly, we show that KTD exhibits asymptotically optimal properties, distinguishing it from the vanilla KI.

3. Our experimental analysis validates the effectiveness of KTD by demonstrating its ability to achieve the desired FDR level with competitive power.

## 2 RELATED WORK

### 2.1 TRAINING DATA LEAKAGE IN LLMS

Memorization in language models, a key aspect of training data leakage, has been widely studied. Research such as Kandpal et al. (2022); Carlini et al. (2021; 2022b); Zeng et al. (2024) examines the memorization behaviors of language models, offering insights into their underlying mechanisms. However, these studies do not propose practical methods for detecting training samples.

In the context of LLMs, other works (Brown et al., 2020; Wei et al., 2021; Du et al., 2022) explore the potential impact of training data leakage on evaluation results. To ensure reliable assessments,

Table 1: Algorithmic properties

| Property | FX-Knockoff | MX-Knockoff | Contamination Test | Mink% | Time Traveling | KTD (Ours) |
|---|---|---|---|---|---|---|
| Metadata-free | ✓ | ✓ | ✓ | ✓ | ✗ | ✓ |
| Threshold-free | ✓ | ✓ | ✓ | ✗ | ✓ | ✓ |
| FDR Control | ✓ | ✓ | ✗ | ✗ | ✗ | ✓ |
| FDR analysis | ✗ | ✗ | ✗ | ✗ | ✗ | ✓ |
| Power analysis | ✗ | ✗ | ✗ | ✗ | ✗ | ✓ |

these studies exclude test samples that share n-gram overlaps with any data used during pre-training. These approaches requires access to pre-training datasets, making it infeasible to detect training samples without support from the model provider.

To evaluate training data leakage without access to the training dataset, various methods have been proposed for detecting training samples. For example, Golchin & Surdeanu (2023) prompt a model to generate completions using two types of instructions and identify training samples by comparing the resulting texts. Shi et al. (2023) assume that non-training texts are more likely to contain outlier tokens that induce significantly higher loss and propose detecting training samples based on top-$k$ token log probabilities. While these methods have shown empirical effectiveness, they do not offer theoretical guarantees for controlling the FDR. Considering this, some studies leverage statistical methods to ensure controlled outcomes. Oren et al. (2023) assume sample exchangeability in non-training datasets and apply hypothesis testing to detect data contamination. Dekoninck et al. (2024) define contamination based on performance differences between the tested and reference models, relying on a predefined threshold $\delta$ to determine significance. However, these methods are mainly designed for dataset-level contamination detection, which differs from our scenario.

Membership inference attacks (MIAs) have also garnered significant attention in the context of training data detection. Similar to our approach, Mattern et al. (2023); Fu et al. (2024) generate neighboring samples that resemble the original ones and compare each sample with its generated neighbors to identify training data. However, these methods do not explicitly aim to control the FDR and rely primarily on empirical observations rather than rigorous theoretical foundations. Carlini et al. (2022a); Mireshghallah et al. (2022), consider FDR as an evaluation metric in their paper. However, these methods are not inherently designed for FDR control and require either training a large number of shadow models or accessing the distribution of data that was not used to train the target model. These constraints make them impractical for LLM scenarios.

## 2.2 KNOCKOFFS

The knockoff framework was first introduced by Barber & Candès (2015) as a data-driven approach to controlling the FDR in variable selection for sparse regression problems. It was later extended to high-dimensional regression by Candes et al. (2018). Over time, knockoff inference has been adapted for various applications, including multi-task regression (Dai & Barber, 2016), outlier detection (Xu et al., 2016), and sample selection (Wang et al., 2024). To the best of our knowledge, we are the first to apply knockoff inference to LLM training data detection.

To highlight the novelty of our proposed method, KTD, we compare it with existing works from both the knockoff and training data detection literature. Specifically, we contrast KTD with FX-Knockoff (Barber & Candès, 2015) and MX-Knockoff (Candes et al., 2018) from the knockoff literature, as well as Contamination Test (Oren et al., 2023), Mink% (Shi et al., 2023), and Time Traveling (Golchin & Surdeanu, 2023) from the training data detection literature, as summarized in Table 1. In the table, **"Metadata-free"** indicates that the method does not require auxiliary information such as dataset names or partition details, while **"Threshold-free"** refers to methods that do not rely on heuristically determined thresholds. This comparison underscores the comprehensiveness of KTD, demonstrating our contributions in both training data detection and knockoff-based inference.

## 3 BACKGROUND

**Notation** We use bold letters to represent vectors of random variables, e.g., $\mathbf{X} = \{X_1, X_2, \ldots, X_n\}$. Furthermore, let $\mathbf{X}_{-j}$ denote the vector resulting from the exclusion of the $j$-th variable $X_j$, i.e., $\mathbf{X} \backslash \{X_j\}$. The independence between two random variables $X$ and $Y$ is symbolized as $X \perp Y$. Let $[n]$ represent the set $\{1, 2, \ldots, n\}$; for any given set $\mathcal{A}$, $|\mathcal{A}|$ denotes the cardinality of $\mathcal{A}$. For two number $a$ and $b$, let $a \vee b$ represent $\max(a, b)$.

**Problem Definition**    Suppose we have $n$ potential training samples $X_1, X_2, \ldots, X_n$ to be tested. For the sake of clarity afterward, we defined $\mathcal{D}_{\text{total}}$ as the index set of these samples, i.e., $\mathcal{D}_{\text{total}} = [n]$. Depending on whether a sample has been used for training the model, $\mathcal{D}_{\text{total}}$ can be divided into two disjoint subsets: $\mathcal{D}_{\text{train}}$ and $\mathcal{D}_{\text{non-train}}$, which represent the index of training samples and non-training samples respectively. Given the dataset $\mathcal{D}_{\text{total}}$ and the language model $f_{\boldsymbol{\theta}}$, our objective is to identify an estimate of $\mathcal{D}_{\text{train}}$, denoted as $\hat{\mathcal{S}} \subset \mathcal{D}_{\text{total}}$, with an FDR bounded by a predefined threshold $q$, while maintaining as high power as possible. Here, the power and FDR are defined as:

$$\text{Power} := \mathbb{E}\left[\frac{|\hat{\mathcal{S}} \cap \mathcal{D}_{\text{train}}|}{|\mathcal{D}_{\text{train}}| \vee 1}\right] \qquad \text{and} \qquad \text{FDR} := \mathbb{E}\left[\frac{|\hat{\mathcal{S}} \cap \mathcal{D}_{\text{non-train}}|}{|\hat{\mathcal{S}}| \vee 1}\right]. \qquad (1)$$

**Auto-regressive LLMs**    The goal of auto-regressive large language models is to capture the underlying language distribution $P_{\boldsymbol{\theta}}(X)$. They achieve this by predicting the next token in a sequence based on the preceding tokens. Specifically, given a sequence $X = (x_1, x_2, \cdots, x_T)$, these models represent the probability of $X$ using the chain rule:

$$P_{\boldsymbol{\theta}}(X) = \prod_{t=1}^{T} P_{\boldsymbol{\theta}}(x_t \mid x_1, x_2, \ldots, x_{t-1})$$

where $\boldsymbol{\theta}$ denotes the parameters of the language model. The parameters $\boldsymbol{\theta}$ are trained to maximize the log-likelihood of sequences in the training dataset.

## 4    METHODOLOGY

In this section, we introduce our knockoff inference-based training data detector, KTD. We begin by illustrating two critical procedures of KTD, which include synthetic knockoff generation and knockoff statistic calculation. Then, the asymptotic analysis for KTD is provided.

### 4.1    KTD: A NOVEL KNOCKOFF-BASED FRAMEWORK

**Motivation**    Knockoff inference is a method originally designed for selecting variables relevant to certain outputs of interest while controlling FDR. In our settings, we can reformulate our problem as a variable selection problem by treating the training samples $\{X_j\}_{j \in \mathcal{D}_{\text{train}}}$ as relevant variables and model parameter $\boldsymbol{\theta}$ as the output of training algorithm $\mathcal{A}lg$ which takes training samples as input, i.e., $\boldsymbol{\theta} = \mathcal{A}lg(\{X_j\}_{j \in \mathcal{D}_{\text{train-total}}})$. Here, $\mathcal{D}_{\text{train-total}}$ represents the model's entire training dataset, and $\mathcal{D}_{\text{train}}$ is a subset of it. Through this reformulation, the robust FDR control ability of KI can be utilized in our context. Since in our approach, we treat samples as random variables, we will use these two terms interchangeably in the following text.

Intuitively, the fundamental idea behind the knockoff inference-based method is to identify relevant variables by comparing them with their noisy counterparts, known as knockoffs. As a result, the knockoff inference-based method usually involves two critical procedures: first, generating knockoffs for the text samples to be tested; second, calculating the knockoff statistic for these text samples by comparing the scores assigned to them with the scores assigned to their knockoff counterparts. Next, we illustrate how these two stages work in vanilla KI and how they are instantiated in KTD.

#### 4.1.1    KNOCKOFF GENERATION

In vanilla KI, knockoffs are typically generated based on specific assumptions about the distribution of variables $\{X_j\}_{j=1}^{n}$, such as Gaussian (Candes et al., 2018), Markov model Sesia et al. (2018) and hidden Markov model (Sesia et al., 2018). However, due to the complexity of natural language, it is challenging to use common distributions to model the relationships between text samples, rendering existing methods unsuitable for knockoff generation in this context. Consequently, we directly adhere to the fundamental definition of knockoffs:

**Definition 1.** *Model-X Knockoffs, (Candes et al., 2018) Model-X knockoffs for a family of random variables $\mathbf{X} = \{X_1, X_2, \ldots, X_n\}$ are a new family of random variables $\tilde{\mathbf{X}} = \{\tilde{X}_1, \tilde{X}_2, \ldots, \tilde{X}_n\}$ satisfying:*

    *1. $\tilde{\mathbf{X}} \perp \boldsymbol{\theta} \mid \mathbf{X}$.*

2. *For any $s \subset [n]$, $(\mathbf{X}, \tilde{\mathbf{X}})_{\text{swap}(s)} \overset{d}{=} (\mathbf{X}, \tilde{\mathbf{X}})$.*

*Here, $(\mathbf{X}, \tilde{\mathbf{X}}) = (X_1, X_2, \ldots, X_n, \tilde{X}_1, \tilde{X}_2, \ldots, \tilde{X}_n)$, and $(\mathbf{X}, \tilde{\mathbf{X}})_{\text{swap}(s)}$ is obtained by swapping $X_j$ with its corresponding knockoff $\tilde{X}_j$ for all $j \in s$. For example, when $n = 3$, $(\mathbf{X}, \tilde{\mathbf{X}})_{\text{swap}(\{1,3\})} = (\tilde{X}_1, X_2, \tilde{X}_3, X_1, \tilde{X}_2, X_3)$.*

This definition guides generating knockoff texts. Property (1) implies that the knockoff text should not be generated by the model being tested, and property (2) requires that the generated knockoff texts be able to replace the original text samples without altering the overall joint distribution. In other words, the knockoff text should convey the same meaning as the original text but in a different manner. To meet these requirements, we generate knockoffs in the KTD framework using a natural language paraphraser, which restates or rephrases the text while preserving its original meaning.

### 4.1.2 KNOCKOFF STATISTIC CALCULATION

**Knockoff Statistic in Vanilla KI**   After constructing the knockoffs, a test statistic known as the knockoff statistic is calculated for each text sample by comparing the importance of the original text sample with that of its knockoff counterparts. This statistic can be viewed as a relevance measure for each text sample and will serve as the basis for training sample selection, which is defined as

**Definition 2.** *Knockoff Statistic, (Candes et al., 2018) A knockoff statistic $\mathbf{W} = \{W_1, W_2, \ldots, W_n\}$ is a measure of variable importance that satisfies the following conditions:*

*1. $\mathbf{W}$ depends only on $\mathbf{X}$, $\tilde{\mathbf{X}}$, and $\boldsymbol{\theta}$:*

$$\mathbf{W} = f(\mathbf{X}, \tilde{\mathbf{X}}, \boldsymbol{\theta}). \tag{2}$$

*2. Swapping the original variable $X_j$ with its corresponding knockoff $\tilde{X}_j$ switches the sign of $W_j$:*

$$W_j([\mathbf{X}, \tilde{\mathbf{X}}]_{\text{swap}(s)}, \boldsymbol{\theta}) = \begin{cases} W_j([\mathbf{X}, \tilde{\mathbf{X}}], \boldsymbol{\theta}), & \text{if } j \notin s \\ -W_j([\mathbf{X}, \tilde{\mathbf{X}}], \boldsymbol{\theta}), & \text{if } j \in s. \end{cases} \tag{3}$$

Typically, the calculation of the knockoff statistic of each variable can be decomposed into two steps. First, assign importance scores $Z_j$ and $\tilde{Z}_j$ to each variable $X_j$ and its knockoff $\tilde{X}_j$ respectively, where the importance scores are calculated by a pre-defined scoring function $T$, i.e.,

$$Z_j = T_j([\mathbf{X}, \tilde{\mathbf{X}}], \boldsymbol{\theta}) \qquad \text{and} \qquad \tilde{Z}_j = T_{j+n}([\mathbf{X}, \tilde{\mathbf{X}}], \boldsymbol{\theta}). \tag{4}$$

Next, calculate the knockoff statistic of $j$-th sample by

$$W_j = Z_j - \tilde{Z}_j. \tag{5}$$

Intuitively, the scores $Z_j$ and $\tilde{Z}_j$ represent the importance of the original sample $X_j$ and its knockoff $\tilde{X}_j$, respectively. A positive $W_j$ ($W_j > 0$) indicates that the $j$-th sample is more relevant to the model parameter $\boldsymbol{\theta}$ than its knockoff, implying its membership in the training data. Conversely, a negative $W_j$ ($W_j < 0$) suggests that the $j$-th sample is more likely to be irrelevant to $\boldsymbol{\theta}$.

During this procedure, the key is to select an appropriate scoring function $t$ that can effectively measure the importance of each sample and its knockoff. In our scenario, we aim to ensure that samples seen by the model are assigned higher scores. Inspired by works using gradient information for OOD detection (Huang et al., 2021; Liang et al., 2018), we use the $L_2$ norm of the model's gradient as the score in KTD, which is defined as:

$$Z_j = -\left\| \frac{\partial \log P_{\boldsymbol{\theta}}(X_j)}{\partial \boldsymbol{\theta}} \right\|_2 \qquad \text{and} \qquad \tilde{Z}_j = -\left\| \frac{\partial \log P_{\boldsymbol{\theta}}(\tilde{X}_j)}{\partial \boldsymbol{\theta}} \right\|_2 \tag{6}$$

where $P_{\boldsymbol{\theta}}(\cdot)$ represents the probability distribution modeled by the model.

Finally, a threshold is determined for thresholding knockoff statistics for training sample detection with an FDR control guarantee. This procedure is illustrated as follows:

**Proposition 1.** *By choosing the threshold $\tau$ according to*

$$\tau = \arg\min_{t>0} \left\{ \frac{1 + |\{j \in [n] : W_j \leq -t\}|}{|\{j \in [n] : W_j \geq t\}| \vee 1} \leq q \right\}. \tag{7}$$

*and setting $\hat{\mathcal{S}} = \{j : W_j \geq \tau\}$, the procedure can control the FDR at $\leq q$.*

**Our Calculation of Knockoff Statistic**   Despite the effectiveness of vanilla KI in controlling the FDR, its knockoff statistic $W_j$ is prone to high variance due to the inherent randomness in the knockoff generation process. This variability makes it more difficult to distinguish between training and non-training samples, leading to the selection of a more conservative threshold $\tau$. Consequently, a larger number of true training samples are excluded to maintain FDR control, resulting in a significant power reduction.

To address this issue, we modify the calculation of the vanilla knockoff statistic $W_j$ by considering multiple draws of the knockoff variables $\tilde{\mathbf{X}}$. Specifically, we calculate the knockoff statistic in KTD as follows:

$$W_j^{\mathrm{KTD}} = Z_j - \frac{1}{m} \sum_{i=1}^{m} \tilde{Z}_j^{(i)} \tag{8}$$

where $\tilde{Z}_j^{(i)}$ is the score calculated based on the $\tilde{X}_j^{(i)}$, the $i$-th draw of $\tilde{X}_j$. Clearly, $W_j$ is a special case of $W_j^{\mathrm{KTD}}$ when $m = 1$. By taking multiple knockoff draws into consideration, we can reduce the variance of knockoff statistic $W_j^{\mathrm{KTD}}$, thereby enhancing the separability between training and non-training samples.

Next, we show that $W_j^{\mathrm{KTD}}$ can also select the appropriate threshold for FDR controlling as $W_j$ do in Proposition 1. We first give the independence assumption of $W_j^{\mathrm{KTD}}$ following Nguyen et al. (2020):

**Assumption 1.** *For any $j \in \mathcal{D}_{\mathrm{non-train}}$, the knockoff statistic $W_j^{\mathrm{KTD}}$ defined in Equation 8 are independent with each other.*

Next, we illustrate the symmetric property of $W_j^{\mathrm{KTD}}$:

**Lemma 1.** $W_j^{\mathrm{KTD}}$ *associated with irrelevant samples is symmetrically distributed around 0, i.e.,*

$$P(W_j^{\mathrm{KTD}} < -t) = P(W_j^{\mathrm{KTD}} > t) \quad \text{for any } t > 0 \text{ and } j \in \mathcal{D}_{\mathrm{non-train}}. \tag{9}$$

This Lemma is empirically validated by our experiments. For details please refer to the third part of our experimental results.

This Lemma, combined with the independence assumed in Assumption 1 implies that the number of non-training samples whose $W_j^{\mathrm{KTD}} > 0$ equals the number of non-training samples whose $W_j^{\mathrm{KTD}} < 0$. This conclusion allows the use of the right-hand side of Equation 10 as an upper bound for FDR, thereby providing an FDR control guarantee.

Consequently, we can select training samples while controlling the FDR using $W_j^{\mathrm{KTD}}$ through a procedure similar to that described in Proposition 1, defined as follows:

**Proposition 2.** *Assume $\{W_j^{\mathrm{KTD}}\}_{j=1}^{n}$ are independent with each other, by choosing the threshold $\tau$ according to*

$$\tau = \min_{t>0} \left\{ \frac{1 + |\{j \in [n] : W_j^{\mathrm{KTD}} \leq -t\}|}{|\{j \in [n] : W_j^{\mathrm{KTD}} \geq t\}| \vee 1} \leq q \right\} \tag{10}$$

*and setting $\hat{\mathcal{S}} = \{j : W_j^{\mathrm{KTD}} \geq \tau\}$, the procedure can control the FDR at $\leq q$.*

## 4.2   The Asymptotic Optimal Property of knockoff statistic in KTD

Here, we provide analysis inspired by Zhao et al. (2022) to illustrate the asymptotic optimality of FDR and power in KTD. We begin by stating an assumption on which these theorems rely.

**Assumption 2.** *For any $j \in \mathcal{D}_{\mathrm{train}}$, we have $\mathbb{E}[W_j^{\mathrm{KTD}}] > 0$.*

**Remark 1.** *This assumption ensures that, on average, training samples will have higher importance scores than their knockoff counterparts. This is reasonable because a sample that has been seen by the model will induce fewer updates (thus higher $Z_j$) compared to its knockoff. Like Lemma 1, we provide the empirical validation of this assumption in our experiment section.*

**Theorem 1.** *Assuming Assumption 2 holds, the variable selection procedure described in Proposition 2 satisfies*

$$\text{Power} = \mathbb{E}\left[ \frac{|\hat{\mathcal{S}} \cap \mathcal{D}_{\mathrm{train}}|}{|\mathcal{D}_{\mathrm{train}}|} \right] \to 1 \quad \text{as} \quad m \to \infty. \tag{11}$$

**Theorem 2.** *Assuming Assumption 2 holds and the threshold $\tau$ found by Proposition 2 is not equal to 0, the variable selection procedure satisfies*

$$\text{FDR} = \mathbb{E}\left[\frac{|\hat{\mathcal{S}} \cap \mathcal{D}_{\text{non-train}}|}{|\hat{\mathcal{S}}|}\right] \to 0 \quad as \quad m \to \infty. \tag{12}$$

**Remark 2.** *Intuitively, as $m$ approaches infinity, the values of $W_j^{\text{KTD}}$ will become increasingly centralized around their expectations. Consequently, given Assumption 2, the $W_j^{\text{KTD}}$ values corresponding to training and non-training samples will form distinct clusters. This separation allows the KTD method to identify a threshold that optimizes both FDR and power.*

## 5 EXPERIMENTS

In this section, we conduct extensive experiments to validate the empirical efficacy of our proposed method. We begin by testing the effectiveness of our method in terms of FDR control. Following that, we empirically validate the symmetric property of our proposed knockoff statistic $W_j^{\text{KTD}}$. Next, we investigate the influence of $m$, the number of knockoff draws, on FDR control performance and the trade-off between power and FDR. Finally, we conduct experiments using Pythia models with different numbers of parameters to examine how model size affects performance. The code for our experiments is available at `https://github.com/huzr1999/KTD`

### 5.1 SETUP

**Baselines** We first select several classic baselines from MIA literature for comparison. Specifically, they include **LOSS** (Yeom et al., 2018), which uses the auto-regressive loss of a sample to determine whether it has been seen during training; **MinK%** (Shi et al., 2023), which takes the average loss of the top k% tokens with the highest loss as the basis for detection and methods that compare a sample's loss to its zlib compression entropy (**Zlib** (Carlini et al., 2021)), its loss after lowercasing (**Lowercase** (Carlini et al., 2021)), and its loss from a smaller reference model (**Ref** (Mireshghallah et al., 2022)). Then, we compare our method with **vanilla KI**, which is an instance of our method when setting $m = 1$.

**Models** We adopt three popular large language models to evaluate our detection algorithm: GPT-2 (137M parameters) (Radford et al., 2019), Pythia (1.4B parameters) (Biderman et al., 2023), and GPT-Neo (1.3B parameters) (Black et al., 2021). Experiments for larger models are available in Appendix D. For baseline Ref, we employ Distilled-GPT2 (Sanh et al., 2019), Pythia-410m, and GPT-Neo-125m as reference models for the three aforementioned main models, respectively. To generate reliable knockoffs for text samples, we use a paraphraser (Vladimir Vorobev, 2023) with the highest downloads on Hugging Face. This paraphraser is based on the T5-base model and fine-tuned with paraphrased texts generated by ChatGPT. Throughout our experiments, we use the model checkpoints provided by Hugging Face[1].

**Dataset** We conduct our experiments on three datasets: **WikiMIA** (Shi et al., 2023) includes texts collected from Wikipedia events. The dataset is separated into two disjoint parts: one corresponding to events happening before 2017 and the other to events happening after 2023. These two parts are used as training samples and non-training samples, respectively. **XSum** (Narayan et al., 2018) includes summaries of BBC news articles. We select the test set of this dataset and randomly separate it into two parts, corresponding to training and non-training samples. **BBC Real Time** (Li et al., 2024b) includes BBC articles from January 2017 to August 2024. Following the process in Shi et al. (2023), we use the articles published in 2017 as training samples and articles published in 2024 as non-training samples. To evaluate our method, we fine-tune the models using the training parts of these datasets while ensuring that the non-training parts remain unseen by the models.

**Computation and Hyperparameters** All the experiments are run with a single NVIDIA Tesla V100 32GB GPU and a 10-core Intel Xeon (Skylake IBRS) CPU. When the model is too large, we use 16-bit quantization to fit the model into GPU memory. Unless explicitly stated, we fix $m = 10$ for all experiments. All codes are implemented with Pytorch (Paszke et al., 2019).

---

[1] https://huggingface.co/

Table 2: Comparison between KTD and baselines. For each dataset and model, if any methods achieve FDR control, the one with the highest power among them will be bolded. Otherwise, the method with the best FDR will be bolded.

| | WikiMIA | | | | | | XSum | | | | | | BBC Real Time | | | | | |
|---|---|---|---|---|---|---|---|---|---|---|---|---|---|---|---|---|---|---|
| | GPT-2 | | Pythia | | GPT-Neo | | GPT-2 | | Pythia | | GPT-Neo | | GPT-2 | | Pythia | | GPT-Neo | |
| | FDR | Power | FDR | Power | FDR | Power | FDR | Power | FDR | Power | FDR | Power | FDR | Power | FDR | Power | FDR | Power |
| LOSS | 0.179 | 0.928 | 0.175 | 0.999 | 0.145 | 0.996 | **0.153** | **0.150** | 0.163 | 0.938 | 0.133 | 0.843 | 0.135 | 0.950 | 0.122 | 0.999 | 0.123 | 0.998 |
| MinK% | 0.193 | 0.232 | 0.193 | 0.987 | 0.120 | 0.749 | 0.313 | 0.024 | 0.165 | 0.060 | 0.203 | 0.293 | 0.245 | 0.279 | 0.103 | 0.958 | 0.130 | 0.964 |
| Zlib | **0.000** | **0.060** | 0.187 | 0.984 | **0.096** | **0.881** | 0.237 | 0.499 | 0.164 | 0.880 | 0.164 | 0.801 | 0.122 | 0.800 | 0.142 | 0.997 | 0.112 | 0.993 |
| Lowercase | 0.484 | 0.995 | 0.485 | 0.991 | 0.487 | 0.983 | 0.570 | 0.096 | 0.697 | 0.019 | 0.654 | 0.081 | 0.807 | 0.005 | 0.495 | 0.016 | 0.671 | 0.017 |
| Ref | 0.326 | 0.835 | **0.167** | **1.000** | 0.181 | 1.000 | 0.172 | 0.487 | 0.140 | 0.999 | 0.114 | 0.990 | 0.182 | 0.553 | 0.169 | 1.000 | 0.126 | 0.996 |
| Vanilla KI | 0.207 | 0.476 | 0.194 | 0.991 | 0.198 | 0.972 | 0.194 | 0.998 | 0.117 | 0.973 | 0.109 | 0.936 | 0.161 | 0.223 | 0.083 | 0.915 | 0.083 | 0.873 |
| KTD (Ours) | 0.197 | 0.869 | 0.230 | 0.998 | 0.193 | 0.958 | 0.238 | 1.000 | **0.109** | **0.995** | **0.101** | **0.990** | **0.101** | 0.349 | **0.071** | **0.980** | **0.067** | **0.973** |

(a) WikiMIA      (b) XSum      (c) BBC Real Time

Figure 1: The FDR control results on three datasets. We vary the FDR bound $q$ from $0.05$ to $0.95$ and calculate its corresponding FDR and power. Each subplot represents results on a dataset and each line in the subplots represents the results of a model. To clearly visualize the bound, we also plot the red line ($y = x$) in each subplots. If a model's FDR is bounded, its corresponding line should be below the red line.

## 5.2 RESULTS

**The Effectiveness of FDR Control with fixed** $q$    We set the FDR bound to $q = 0.1$ and present a comparative analysis of our method against the baselines in Table 2. Since the baseline methods—LOSS, MinK%, Zlib, and Lowercase—only produce confidence scores for training data membership inference, they require a validation set to determine an appropriate threshold. To accommodate this, we sample a validation set of 100 instances and select thresholds that achieve the highest power while maintaining a bounded FDR on the validation set. For the Ref baseline, which relies on a "general distribution" to determine the threshold, we approximate this distribution by combining all non-training samples from the three datasets. From Table 2, we observe that our method, KTD, effectively controls the FDR on the XSum and BBC Real-Time datasets while maintaining relatively stable performance across different datasets and models. In contrast, baselines such as Zlib, MinK%, and Lowercase occasionally exhibit extremely poor FDR or power, suggesting that their effectiveness is highly dependent on the choice of the validation set. Although some baselines achieve more favorable results on WikiMIA, we argue that this comparison is not entirely fair. These baselines require access to a validation set with ground truth membership labels or an accurate distribution of non-training samples—resources that may not be available in real-world scenarios.

**The Effectiveness of FDR Control under Varying** $q$    We vary $q$ and plot the corresponding power and FDR in Figure 1. The results show that our method effectively controls the FDR in most cases. Although in certain cases (e.g., three models on WikiMIA), the FDR is not strictly bounded by $q$, the corresponding curves closely track the red line, indicating that the FDR remains within $q$ plus a small constant. For a more detailed analysis of these cases please refer to Appendix B.

**The Symmetric Property of** $W_j^{\mathrm{KTD}}$    To empirically validate Lemma 1 and Assumption 2, we plot the distribution of $W_j^{\mathrm{KTD}}$ calculated on the BBC Real Time dataset in Figure 2. From the figure, we can make three observations. Firstly, the $W_j^{\mathrm{KTD}}$ of non-training samples are symmetrically distributed around 0, which is aligned with Lemma 1. Secondly, the expectation of non-training samples' $W_j^{\mathrm{KTD}}$ is greater than 0, illustrating the validity of Assumption 2, which Theorems 1 and 2

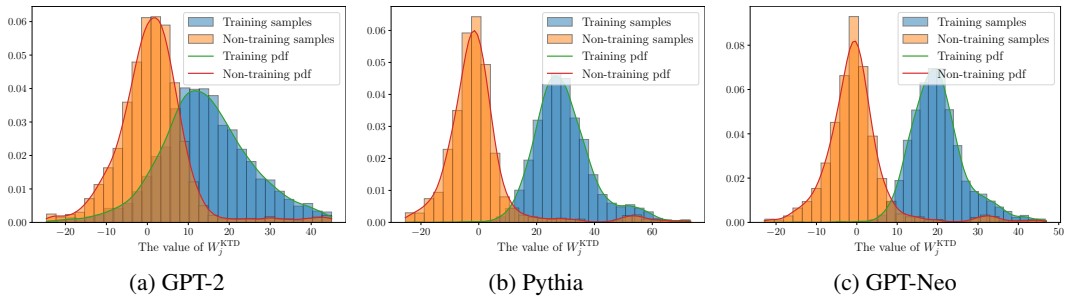

Figure 2: The distribution of our knockoff statistic $W_j^{\text{KTD}}$.

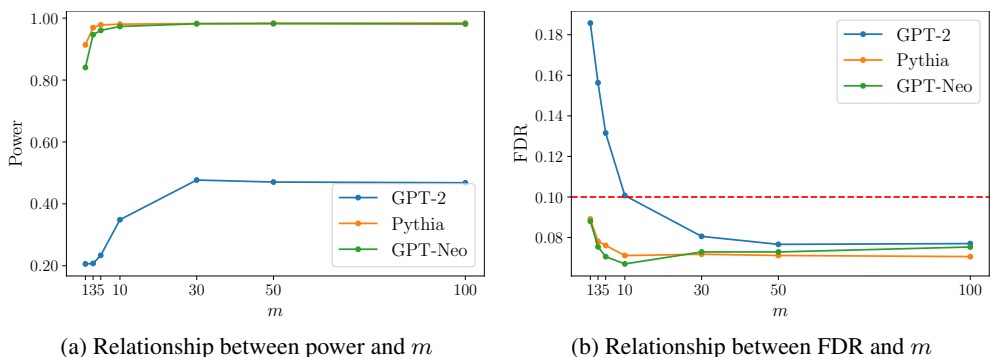

(a) Relationship between power and $m$      (b) Relationship between FDR and $m$

Figure 3: $m$'s influence on power and FDR. We fix the bound $q$ at 0.1 and vary $m$ in $\{1, 3, 5, 10, 30, 50, 100\}$ to calculate the power and FDR for each model. The red line in the figure represents the FDR bound given by $q$. Dots below the red line indicate successful bounding.

rely on. Lastly, larger models (Pythia and GPT-Neo) exhibit better separability between training and non-training samples. This is consistent with expectations, as larger models have a higher probability of memorizing their training data (Carlini et al., 2022b), thereby showing different behavior for training versus non-training samples.

**The Influence of the Number of Knockoff Draws** In KTD, we use multiple knockoff draws to compute the KTD statistic $W_j^{\text{KTD}}$. Here, we justify this design by examining two key questions: (1) how $m$ affects FDR control and power, and (2) how $m$ influences the power-FDR trade-off. Note that vanilla KI is a special case of KTD with $m = 1$.

For the first question, we plot power and FDR against $m$ in Figures 3a and 3b, respectively. Several observations emerge from these figures. First, KTD provides better FDR control than vanilla KI. Specifically, vanilla KI fails to control the FDR for GPT-2, whereas KTD successfully bounds it when $m \geq 10$. Second, increasing $m$ benefits both FDR control and power—both metrics improve consistently as $m$ grows. Finally, for Pythia and GPT-Neo, power approaches 1 while FDR approaches 0 as m increases, validating the asymptotic optimality stated in Theorems 1 and 2.

To address the second question, we plot power-FDR pareto curves for different $m$ values in Figure 4. The figure shows that as $m$ increases, the curves shift toward the upper left corner, indicating an improved power-FDR trade-off. Even small values, such as $m = 3$, yield significant improvements. This demonstrates that multiple knockoff draws substantially enhance the balance between power and FDR. Additionally, for GPT-2, we observe an unusual trend when $m$ is small (1 to 5): sometimes, FDR decreases as the target FDR level $q$ increases. We hypothesize that this is due to the high instability associated with small $m$.

Since computation time is primarily dominated by the gradient norm calculations for knockoffs, Figure 3 also illustrates the trade-off between computation time and performance. From the figure, we find that $m = 10$ strikes an optimal balance, achieving strong performance within a reasonable computation time.

**The influence of Model Size** We evaluate the FDR in terms of the power-FDR trade-off across three datasets using different-sized Pythia models (440M, 1B, 1.4B, 2.8B). Figure 5 shows a clear

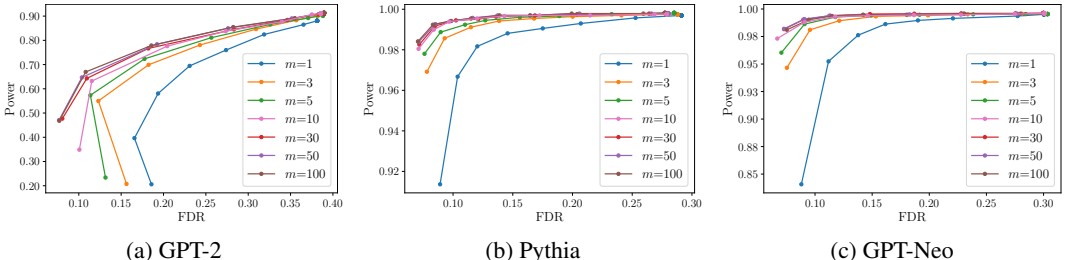

(a) GPT-2          (b) Pythia          (c) GPT-Neo

Figure 4: Trade-off between FDR and power under different $m$. For each $m$, we vary the FDR bound $q$ and calculate the corresponding trade-off between power and FDR. Each subplot represents a model and each line in the subplots represents the trade-off curve under certain $m$. The closer the curve is to the upper left corner, the better the trade-off between power and FDR.

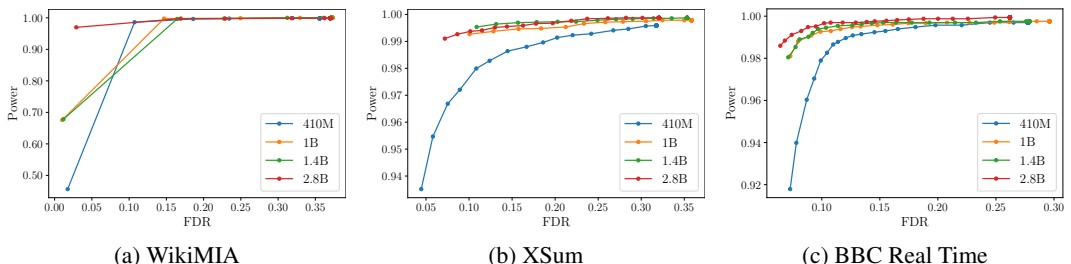

(a) WikiMIA          (b) XSum          (c) BBC Real Time

Figure 5: Trade-off between FDR and power under different model sizes. For each model size, varying FDR bound $q$ is applied to compute the trade-off between power and FDR. Each subplot presents the results on a different dataset, with each line representing the trade-off curve of a model. Curves closer to the upper-left corner indicate a more favorable balance between power and FDR.

trend that the trade-off improves as model size increases. This aligns with our expectations, as larger models are more likely to memorize training data, making it easier to distinguish between training and non-training samples. Moreover, models with sizes larger than 1 billion parameters exhibit relatively high power even when a strict FDR bound is imposed. This observation suggests that 1 billion parameters may serve as a threshold, beyond which models can easily memorize samples from these three datasets, thereby making the distinction between training and non-training samples exceptionally clear.

## 6 CONCLUSION

In this paper, we tackled the critical issue of detecting training data for LLMs with a focus on controlling FDR and introduced a novel knockoff-based method, KTD. KTD instantiates the KI framework in the context of training data detection and employs a novel calculation method that leverages multiple knockoff draws to address the high variance of the knockoff statistic in vanilla KI. To support KTD, we provided theoretical guarantees for KTD, demonstrating that it not only effectively controls the FDR but also possesses asymptotic optimal properties. Our empirical evaluations on three datasets further validated the efficacy of KTD, showcasing its superior performance in terms of FDR control and the power-FDR trade-off compared to existing methods.

## 7 LIMITATIONS

The limitations of our method primarily stem from two factors. First, the effectiveness of our approach depends on access to the gradients of LLMs, which may not be available for certain proprietary models where gradient information is inaccessible. This dependency also limits the applicability of our method to tasks such as paraphrased text detection. Second, our method assumes the availability of a high-quality paraphraser to generate knockoff samples. This reliance on paraphraser quality introduces a potential bottleneck in achieving optimal performance. In the future, we will explore the possibility of designing a framework for FDR control that relies solely on logits or even the output text of LLMs.

## 8 ACKNOWLEDGMENTS

This project is supported by the National Research Foundation, Singapore, under its NRF Professorship Award No. NRF-P2024-001.

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

# A  APPENDIX

## A.1  THE PROOF OF LEMMA 1

**Lemma 1.** $W_j^{\mathrm{KTD}}$ *associated with irrelevant samples is symmetrically distributed around 0, i.e.,*

$$P(W_j^{\mathrm{KTD}} < -t) = P(W_j^{\mathrm{KTD}} > t) \quad \text{for any } t > 0 \text{ and } j \in \mathcal{D}_{\mathrm{non-train}}. \tag{13}$$

*Proof.* For simplicity, let $W_j^{(i)}$ denote $Z_j - Z_j^{(i)}$. Then,

$$
\begin{aligned}
P(W_j^{\mathrm{KTD}} < -t) &= P\left(\frac{1}{m}\sum_{i=1}^{m} W_j^{(i)} < -t\right) \\
&= \int_{-\infty}^{-t}\int_{D_v} P(W_j^{(1)} = a_1, W_j^{(2)} = a_2, \dots, W_j^{(m)} = a_m)\, da_1\, da_2\, \dots\, da_m\, dv,
\end{aligned}
\tag{14}
$$

where $D_v = \{(a_1, a_2, \dots, a_m) \mid \frac{1}{m}\sum_{i=1}^{m} a_i = v\}$.

According to Lemma 3.3 in Candes et al. (2018), the signs of $W_j^{(i)}$ are independent of their magnitudes. As a result, we can express the probability $P(W_j^{(1)} = a_1, W_j^{(2)} = a_2, \dots, W_j^{(m)} = a_m)$ as the product of the following two terms:

1. $P(|W_j^{(1)}| = |a_1|, |W_j^{(2)}| = |a_2|, \dots, |W_j^{(m)}| = |a_m|)$,

2. $\prod_{i=1}^{m} P(\mathrm{sign}(W_j^{(i)}) = \epsilon_i)$, where $\epsilon_i = \mathrm{sign}(a_i) \in \{-1, 1\}$.

Given the symmetric property of standard knockoff statistic $W_j$, we have $P(\mathrm{sign}(W_j^{(i)}) = -1) = P(\mathrm{sign}(W_j^{(i)}) = 1)$. Therefore, switching the sign of $W_j^{(i)}$ will not affect the probability above. Consequently, for any element $(a_1, a_2, \dots, a_m)$ in $D_v$, there exists a corresponding element $(-a_1, -a_2, \dots, -a_m)$ in $D'_v = \{(a'_1, a'_2, \dots, a'_m) \mid \frac{1}{m}\sum_{i=1}^{m} a'_i = -v\}$ satisfying $P(W_j^{(1)} = a_1, W_j^{(2)} = a_2, \dots, W_j^{(m)} = a_m) = P(W_j^{(1)} = a'_1, W_j^{(2)} = a'_2, \dots, W_j^{(m)} = a'_m)$.

As a result, Equation 14 equals:

$$
\int_{-\infty}^{-t} \int_{D_v} P(W_j^{(1)} = a_1, W_j^{(2)} = a_2, \ldots, W_j^{(m)} = a_m)\, da_1\, da_2\, \ldots\, da_m\, dv
$$

$$
= \int_{-\infty}^{-t} \int_{D_v'} P(W_j^{(1)} = a_1', W_j^{(2)} = a_2', \ldots, W_j^{(m)} = a_m')\, da_1\, da_2\, \ldots\, da_m\, dv
$$

$$
= \int_{t}^{\infty} \int_{D_v} P(W_j^{(1)} = a_1, W_j^{(2)} = a_2, \ldots, W_j^{(m)} = a_m)\, da_1\, da_2\, \ldots\, da_m\, dv \tag{15}
$$

$$
= P\left( \frac{1}{m} \sum_{i=1}^{m} W_j^{(i)} > t \right)
$$

$$
= P(W_j^{\mathrm{KTD}} > t).
$$

$\square$

## A.2 THE PROOF OF THEOREM 1

**Theorem 1.** *Assuming Assumption 2 holds, the variable selection procedure described in Proposition 2 satisfies*

$$
\mathrm{Power} = \mathbb{E}\left[ \frac{|\hat{\mathcal{S}} \cap \mathcal{D}_{\mathrm{train}}|}{|\mathcal{D}_{\mathrm{train}}|} \right] \to 1 \quad as \quad m \to \infty. \tag{16}
$$

*Proof.* The main idea of the proof comes from (Zhao et al. (2022), Theorem 6). For the sake of clarity of the article, we provide the complete proof here. Let $\xi_j$ denote the expectation of $W_j^{\mathrm{KTD}}$, i.e., $\xi_j = \mathbb{E}[W_j^{\mathrm{KTD}}]$, and let $\xi$ be the minimum $\xi_j$ among the relevant variable set $\mathcal{D}_{\mathrm{train}}$, i.e., $\xi = \min_{j \in \mathcal{D}_{\mathrm{train}}} \xi_j$. Use $\sigma_j$ to represent the standard error of $W_j^{\mathrm{KTD}}$.

For any $j \in \mathcal{D}_{\mathrm{train}}$, we have

$$
\begin{aligned}
P\left( W_j^{\mathrm{KTD}} > -\frac{\xi}{2} \right) &\geq P\left( W_j^{\mathrm{KTD}} > \frac{\xi_j}{2} \right) \\
&\geq P\left( |W_j^{\mathrm{KTD}} - \xi_j| < \frac{\xi_j}{2} \right) \\
&\geq 1 - \frac{4\sigma_j^2}{\xi_j^2 m} \\
&\geq 1 - \frac{4\sigma_j^2}{\xi^2 m}.
\end{aligned} \tag{17}
$$

Due to the symmetric property of elements in $\mathcal{D}_{\mathrm{non-train}}$, we have $\xi_j = 0$ for all $j \in \mathcal{D}_{\mathrm{non-train}}$. Hence, similar to the above equation, the following formula holds:

$$
P\left( W_j^{\mathrm{KTD}} > -\frac{\xi}{2} \right) \geq 1 - \frac{4\sigma_j^2}{\xi^2 m} \qquad \forall j \in \mathcal{D}_{\mathrm{non-train}}. \tag{18}
$$

Combining Equation 17 and Equation 18, we get:

$$
\begin{aligned}
P\left( \min_j W_j^{\mathrm{KTD}} < -\frac{\xi}{2} \right) &= P\left( W_1^{\mathrm{KTD}} < -\frac{\xi}{2} \vee W_2^{\mathrm{KTD}} < -\frac{\xi}{2} \vee \cdots \vee W_n^{\mathrm{KTD}} < -\frac{\xi}{2} \right) \\
&= \sum_{j=1}^{n} P\left( W_j^{\mathrm{KTD}} < -\frac{\xi}{2} \right) \\
&\leq \frac{4}{\xi^2 m} \sum_{j=1}^{n} \sigma_j^2.
\end{aligned} \tag{19}
$$

If $\min_j W_j^{\text{KTD}} < -\frac{\xi}{2}$, according to the procedure described in Proposition 2 to determine the threshold, we have $\tau \leq \max\{0, -\min_j W_j^{\text{KTD}}\}$. Therefore, we can derive a lower bound for the power:

$$\text{Power} = \mathbb{E}\left[\frac{|\mathcal{D}_{\text{train}} \cap \hat{\mathcal{S}}|}{|\mathcal{D}_{\text{train}}|}\right] \geq \mathbb{E}\left[\frac{|\mathcal{D}_{\text{train}} \cap \hat{\mathcal{S}}|}{|\mathcal{D}_{\text{train}}|} \,\middle|\, \min_j W_j^{\text{KTD}} > -\frac{\xi}{2}\right] \cdot P\left(\min_j W_j^{\text{KTD}} > -\frac{\xi}{2}\right). \tag{20}$$

Note that $\tau < \frac{\xi}{2}$. Thus, the above formula is less than or equal to

$$\frac{1}{|\mathcal{D}_{\text{train}}|} \sum_{j \in \mathcal{D}_{\text{train}}} \mathbf{1}\left[W_j^{\text{KTD}} > \frac{\xi}{2}\right] \cdot P\left(\min_j W_j^{\text{KTD}} > -\frac{\xi}{2}\right). \tag{21}$$

This lower bound approaches 1 as $m \to \infty$. $\qquad\square$

## A.3 THE PROOF OF THEOREM 2

**Theorem 2.** *Assuming Assumption 2 holds and the threshold $\tau$ found by Proposition 2 is not equal to 0, the variable selection procedure satisfies*

$$\text{FDR} = \mathbb{E}\left[\frac{|\tilde{\mathcal{S}} \cap \mathcal{D}_{\text{train}}|}{|\hat{\mathcal{S}}|}\right] \to 0 \quad as \quad m \to \infty. \tag{22}$$

*Proof.* Similar to Equation 19, we have:

$$P\left(\min_{j \in \mathcal{D}_{\text{train}}} W_j^{\text{KTD}} < \frac{\xi}{2}\right) \leq \frac{4}{\xi^2 m} \sum_{j \in \mathcal{D}_{\text{train}}} \sigma_j^2. \tag{23}$$

This probability approaches 0 as $m \to \infty$.

We then consider FDR:

$$\mathbb{E}\left[\frac{|\mathcal{D}_{\text{non-train}} \cap \hat{\mathcal{S}}|}{|\hat{\mathcal{S}}|}\right] = \mathbb{E}\left[\frac{|\mathcal{D}_{\text{non-train}} \cap \hat{\mathcal{S}}|}{|\hat{\mathcal{S}}|} \,\middle|\, \min_j W_j^{\text{KTD}} > -\frac{\xi}{2}\right] \cdot P(\min_j W_j^{\text{KTD}} > -\frac{\xi}{2})$$

$$+ \mathbb{E}\left[\frac{|\mathcal{D}_{\text{non-train}} \cap \hat{\mathcal{S}}|}{|\hat{\mathcal{S}}|} \,\middle|\, \min_j W_j^{\text{KTD}} > -\frac{\xi}{2}, \min_{j \in \mathcal{D}_{\text{train}}} W_j^{\text{KTD}} < \frac{\xi}{2}\right] \cdot P(\min_j W_j^{\text{KTD}} > -\frac{\xi}{2}, \min_{j \in \mathcal{D}_{\text{train}}} W_j^{\text{KTD}} < \frac{\xi}{2})$$

$$+ \mathbb{E}\left[\frac{|\mathcal{D}_{\text{non-train}} \cap \hat{\mathcal{S}}|}{|\hat{\mathcal{S}}|} \,\middle|\, \min_j W_j^{\text{KTD}} > -\frac{\xi}{2}, \min_{j \in \mathcal{D}_{\text{train}}} W_j^{\text{KTD}} > \frac{\xi}{2}\right] \cdot P(\min_j W_j^{\text{KTD}} > -\frac{\xi}{2}, \min_{j \in \mathcal{D}_{\text{train}}} W_j^{\text{KTD}} > \frac{\xi}{2}). \tag{24}$$

The probabilities in the first and second terms go to 0 as $m \to \infty$. Hence, we focus on the expectation in the third term. Under the condition $\min_j W_j^{\text{KTD}} > -\frac{\xi}{2}$, we have $\tau < \frac{\xi}{2}$. Additionally, considering $\min_{j \in \mathcal{D}_{\text{train}}} W_j^{\text{KTD}} > \frac{\xi}{2}$, it follows that $|\hat{\mathcal{S}}| > |\mathcal{D}_{\text{train}}|$.

$$\mathbb{E}\left[\frac{|\mathcal{D}_{\text{non-train}} \cap \hat{\mathcal{S}}|}{|\hat{\mathcal{S}}|} \,\middle|\, \min_j W_j^{\text{KTD}} > -\frac{\xi}{2}, \min_{j \in \mathcal{D}_{\text{train}}} W_j^{\text{KTD}} > \frac{\xi}{2}\right]$$

$$\leq \frac{1}{|\mathcal{D}_{\text{train}}|} \sum_{j \in \mathcal{D}_{\text{non-train}}} \mathbf{1}\left[W_j^{\text{KTD}} > \tau\right]. \tag{25}$$

Since we assume $\tau > 0$, there exists a $\delta > 0$ such that $\tau > \delta > 0$. Therefore, the above formula is less than or equal to:

$$\frac{1}{|\mathcal{D}_{\text{train}}|} \sum_{j \in \mathcal{D}_{\text{non-train}}} \frac{\sigma_j}{m\delta}, \tag{26}$$

which goes to 0 as $m \to \infty$. $\qquad\square$

### A.4 PARAMETER SETTINGS

For fine-tuning, we used the following settings:

- warmup_step = 100
- weight_decay = 0.01
- batch_size = 8
- num_epochs = 3 (10 for GPT-2)

All other hyperparameters were set to the default values provided by the 'TrainingArguments' class in the Transformers library.

For paraphrasing, we applied the following configurations:

- Top-k sampling with 'topk' = 50
- Top-p sampling with 'topp' = 0.95
- Temperature scaling with 'temperature' = 1.9

## B EXPERIMENTS ON "GOOD KNOCKOFFS"

### B.1 WHY "GOOD KNOCKOFFS" MATTER?

As shown in Figure 1, TKD cannot strictly control the FDR in the low-FDR region. According to our analysis, the performance limitations observed for small $q$ values are primarily attributed to imperfections in the generated knockoffs.

Theoretically, we prove that the $W_j^{\mathrm{KTD}}$ for non-training samples should be symmetrically distributed around zero (Lemma 1). This proof relies on the assumption that the generated knockoffs are perfectly swappable with the original texts (the second requirement in Definition 1).

However, in natural language processing scenarios, generating perfect knockoffs that fully meet the second requirement in Definition 1 for all texts is inherently challenging, which differs from traditional settings where data distributions are often assumed to follow well-defined distributions, such as the Gaussian distribution. As a result, we observe in our experiments that the distribution of $W_j^{\mathrm{KTD}}$ values for non-training samples is not perfectly symmetrical in some cases.

Next, we illustrate why this asymmetry can impair FDR control. In Equation 10, we use the expression (denoted as $\mathrm{frac}_1(t)$)

$$\frac{1 + |\{j \in [n] : W_j^{\mathrm{KTD}} \leq -t\}|}{|\{j \in [n] : W_j^{\mathrm{KTD}} \geq t\}| \vee 1}$$

(from the left-hand side of Equation 10) to upper bound the true $\mathrm{FDR}(t)$. Here, $|\{j \in [n] : W_j^{\mathrm{KTD}} \leq -t\}|$ acts as an upper bound for the numerator in the definition of $\mathrm{FDR}(t)$, i.e.,

$$\underbrace{|\{j \in [n] : W_j^{\mathrm{KTD}} \leq -t\}|}_{\text{part1}} \geq |\{j \in \mathcal{D}_{\mathrm{non-train}} : W_j^{\mathrm{KTD}} < -t\}| = \underbrace{|\{j \in \mathcal{D}_{\mathrm{non-train}} : W_j^{\mathrm{KTD}} > t\}|}_{\text{part2}},$$
(27)

similar to Equation 3.9 of Candes et al. (2018). Consequently, the violation of the symmetry property results in a situation where $\mathrm{frac}_1(t)$ can no longer serve as a strict upper bound for $\mathrm{FDR}(t)$.

This issue becomes more pronounced as $q$ approaches very small values, since in such cases, the threshold $\tau$ determined by Equation 10 increases. A larger $\tau$ reduces the denominator of $\mathrm{frac}_1(\tau)$ (which is also the denominator of $\mathrm{FDR}(\tau)$), thereby amplifying the impact of the asymmetry-induced disparity between part 1 and part 2 in Equation 27 (the numerators of $\mathrm{frac}_1(\tau)$ and $\mathrm{FDR}(\tau)$). As a result, when $q$ is very small, it becomes increasingly difficult to impose a strict FDR bound.

This highlights the importance of generating high-quality knockoffs, which is especially critical for effective FDR control in low-FDR settings.

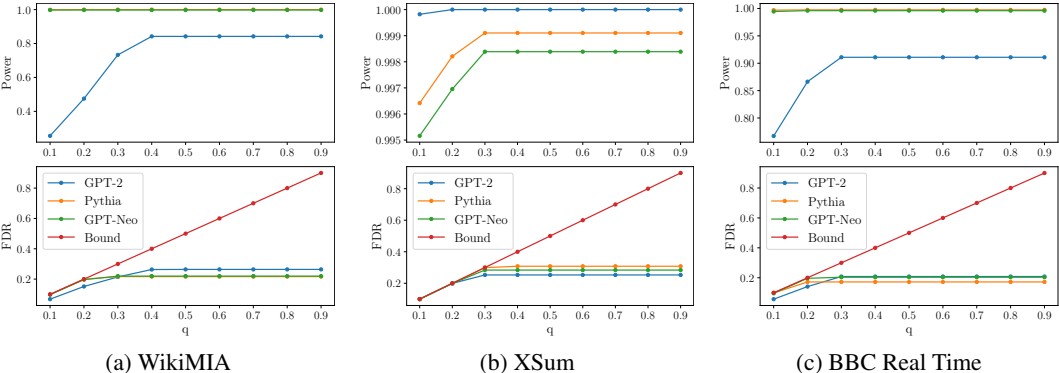

Figure 6: The FDR control results on three datasets with the more symmetrical distribution of $W_j^{\mathrm{KTD}}$. Other settings are identical to those in Figure 1

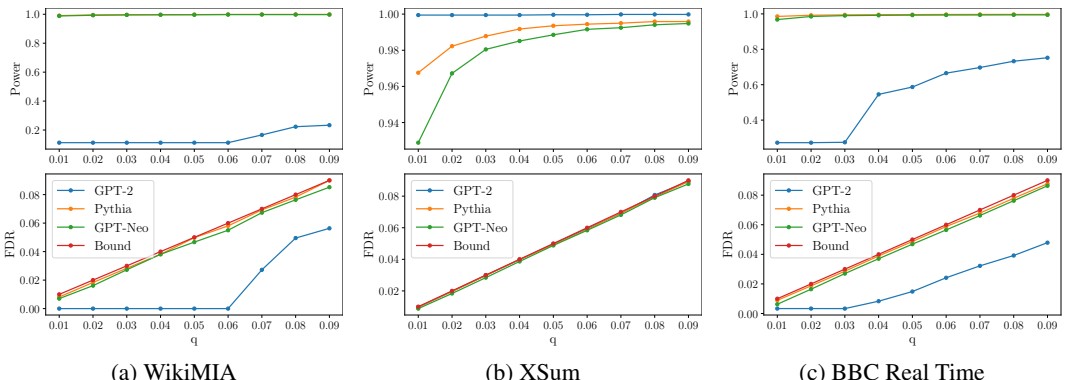

Figure 7: The FDR control results in extremely low FDR-region on three datasets with more symmetrical distribution of $W_j^{\mathrm{KTD}}$. Other settings are identical to those in Figure 1.

### B.2 KTD'S PERFORMANCE WITH GOOD KNOCKOFFS

While the imperfections in the generated knockoffs can impact the effectiveness of our method, particularly in the low-FDR region, we emphasize that the primary focus of this paper is on designing the overall framework rather than optimizing the generation of high-quality knockoffs. We argue that KTD can achieve improved performance with better knockoffs and conduct experiments to empirically illustrate this. In this section, we select a subset of non-training samples to adjust the distribution of their $W_j^{\mathrm{KTD}}$, ensuring greater symmetry. Such selection can be challenging in practical settings, as it is unclear whether each sample is from the training or non-training data. However, our goal here is to explore the performance of KTD under good knockoffs, so we make this ideal selection.

**FDR Controlling Performance** We first evaluate the performance of KTD in terms of FDR control when the distribution of non-training samples' $W_j^{\mathrm{KTD}}$ is made more symmetrical. To be specific, we apply our method in both the regular-FDR region (0.1–0.9) and the extremely low-FDR region (0.01–0.09). The results of these experiments are shown in Figures 6 and 7. From these figures, we observe that a more symmetrical distribution of $W_j^{\mathrm{KTD}}$ for non-training samples enables more strict FDR control, even when the restrain $q$ is extremely low.

**Power-FDR Tradeoff** We extend Figure 4 to explore the extremely low-FDR region, with the results shown in Figure 8. From the figure, we can observe that with a more symmetrical $W_j^{\mathrm{KTD}}$ distribution, the trends in the low-FDR region are consistent with those observed in the ordinary scenario (where FDR > 0.1).

**Robustness to Imbalanced Test Set** We evaluated the performance of KTD in scenarios with highly imbalanced ratios of training to non-training samples in the test set. Specifically, we sampled

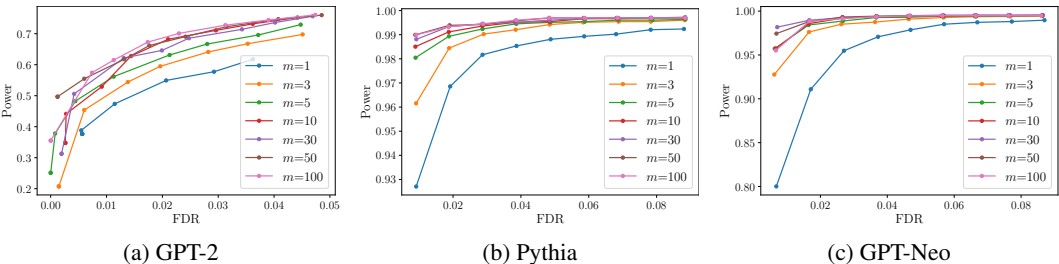

Figure 8: Trade-off between FDR and power under different $m$ in the extremely low-FDR region with more symmetrical distribution of $W_j^{\mathrm{KTD}}$. All other settings are identical to those in Figure 4.

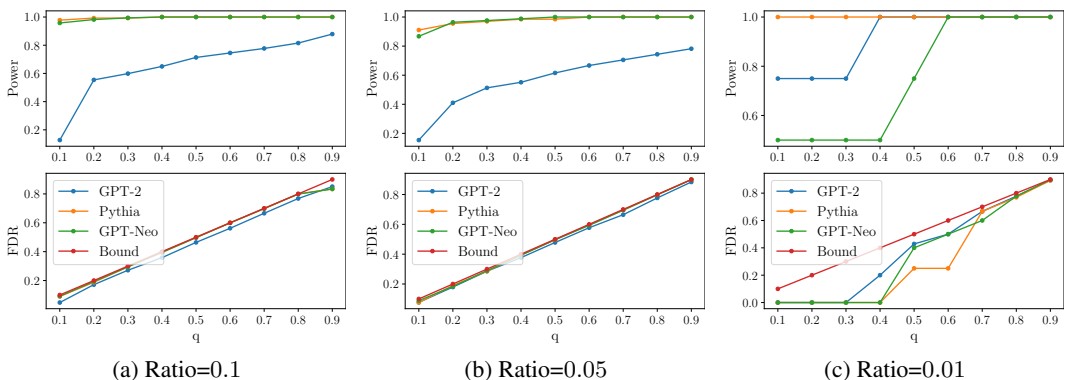

Figure 9: The FDR control results for different training-to-non-training ratios with more symmetrical distribution of $W_j^{\mathrm{KTD}}$. Other settings are identical to those in Figure 1

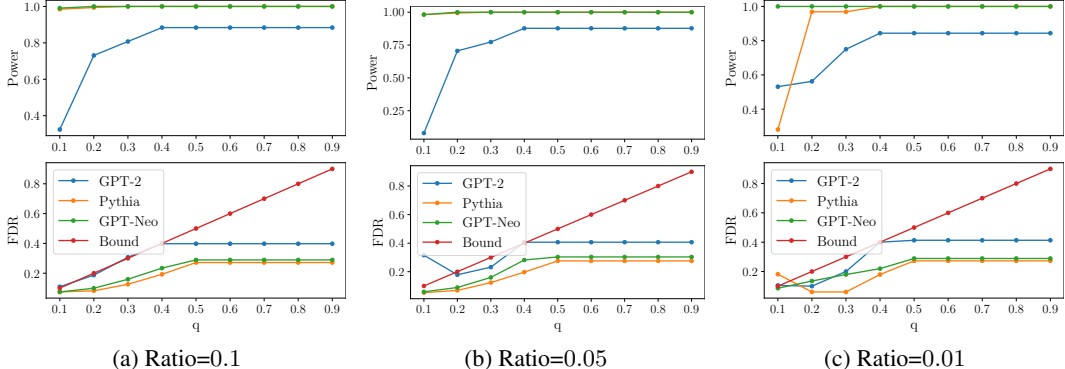

Figure 10: The FDR control results for different diluting degrees. The "Ratio" represents the proportion of training samples selected for testing. Other settings are identical to those in Figure 1

the training data within the test set to achieve training-to-non-training ratios of 0.1, 0.05, and 0.01 in the test set. The results are presented in Figure 9. As shown in the figure, when the $W_j^{\mathrm{KTD}}$ distribution becomes more balanced, KTD demonstrates robustness in highly imbalanced scenarios.

## C  EXPERIMENTS ON DILUTION

In this section, we investigate the robustness of KTD in a "diluted scenario," where the models' fine-tuning data are mixed with "background" samples that are absent from the test set. To simulate this scenario, we use only a proportion of the fine-tuning data for subsequent testing and present the results for different proportions in Figure 10. As shown in the figure, the results follow a trend similar to those in the standard scenario, demonstrating that KTD remains robust across varying levels of dilution.

Table 3: KTD's performance on 7B LLM under varying $q$ (corresponding to Figure 1)

| | q | 0.1 | 0.2 | 0.3 | 0.4 | 0.5 | 0.6 | 0.7 | 0.8 | 0.9 |
|---|---|---|---|---|---|---|---|---|---|---|
| Pythia-6.9B | **Power** | 0.999 | 1.000 | 1.000 | 1.000 | 1.000 | 1.000 | 1.000 | 1.000 | 1.000 |
| | **FDR** | 0.103 | 0.215 | 0.326 | 0.347 | 0.347 | 0.347 | 0.347 | 0.347 | 0.347 |
| Mistral-7B | **Power** | 0.811 | 0.928 | 0.954 | 0.969 | 0.971 | 0.971 | 0.971 | 0.971 | 0.971 |
| | **FDR** | 0.077 | 0.129 | 0.199 | 0.308 | 0.322 | 0.322 | 0.322 | 0.322 | 0.322 |

Table 4: KTD's performance on the 7B model with varying $m$ (corresponding to Figure 3).

| | **Pythia-6.9B** | | | | **Mistral-7B** | | | |
|---|---|---|---|---|---|---|---|---|
| m | 1 | 3 | 5 | 10 | 1 | 3 | 5 | 10 |
| **Power** | 0.877 | 0.945 | 0.957 | 0.967 | 0.520 | 0.735 | 0.778 | 0.811 |
| **FDR** | 0.121 | 0.111 | 0.112 | 0.103 | 0.095 | 0.078 | 0.076 | 0.077 |

# D   EXPERIMENTS ON 7B MODELS

To assess the effectiveness of KTD on models with approximately 7 billion parameters, we present the results of 2 models (Pythia-6.9B and Mistral-7B) on the BBC Real-Time dataset in Tables 3 and 4, corresponding to Figures 1 and 3, respectively. From these tables, we observe that the performance trend of these models is consistent with that of smaller models.

# E   COMPARISON WITH CONSTAT

We attempt to extend Constat Dekoninck et al. (2024) from benchmark-level detection to sample-level detection by treating each sample as an individual benchmark. However, we find that this approach is not well-suited to our setting for the following reasons:

- **Computational complexity**: Using the default number of bootstrapping, evaluating a dataset with approximately 1,400 samples requires around 42 hours, making the approach computationally expensive and impractical in our settings.

- **Choice of predefined** $\delta$: Constat requires users to specify a threshold $\delta$ to determine whether the performance difference between the test model and the reference models is significant. However, in our setting, the range of the gradient norms varies significantly across different datasets and models, making it challenging to select an appropriate $\delta$. We observe that an improperly chosen $\delta$ can lead to trivial results, such as classifying all samples as training samples.

