# OpenReview forum: "A Statistical Approach for Controlled Training Data Detection"
_ICLR.cc/2025/Conference — ICLR 2025 Poster_

### Official Review · Reviewer_Lbdc · 2024-10-21

**Soundness:** 3
**Presentation:** 3
**Contribution:** 3
**Rating:** 6
**Confidence:** 3

**Summary:**

In this study, the authors address the challenge of identifying whether a given sample is part of the training set. Their approach is based on comparing specific KTD scores between perturbed versions of the text and the original, drawing inspiration from the knockoff inference method. The proposed technique is tested on a few smaller models and demonstrated to be effective in certain cases.

**Strengths:**

First, the problem of detecting whether a certain sample is in the training set is an interesting and important problem.

Second, the idea of applying the knockoff Inference method and extending it with multiple samples seems to be a reasonable one.

**Weaknesses:**

First, while I appreciate the authors’ effort in trying to provide some theoretical guarantee on the result, it is apparent to me after reading that multiple steps of the approach remain ad hoc and thus the guarantee actually means little. For instance, the choice of using “the L2 norm of the model’s gradient as the score in KTD”, which is rather important in your method, must be properly justified. That is, why is it a meaningful choice and how do users understand the guarantee based such as scores then? Furthermore, how do you determine the number of samples? Can you provide some statistical analysis for determining the number of samples (e.g., in the spirit of hypothesis testing for instance)?

Secondly, the experimental evaluation can be improved. For instance, can you conduct some experiments to show the effect of different KTD scores? (Intuitively, one may consider choices such as influence - a near zero influence of a certain sample would imply it is not in the training set). Furthermore, can you conduct the experiments on even larger models such as Llama3?

The following are a list of detailed comments.

Page 1: “Detecting training data for large language models (LLMs) is attracting growing attention, particularly in applications that require high reliability. While numerous endeavors have been made to tackle this issue, ensuring controlled results often remains overlooked.”

Comment: I have trouble inferring what is the problem to be solved from this sentence.

Page 1: “To address this issue, recent studies have focused on detecting training data from large language models (LLMs) …”

Comment: The short-hand “LLMs” have already been introduced.

Page 2: “While the knockoff statistic from the vanilla KI method can be directly applied to KTD, it has a significant drawback …”

Comment: You should introduce what the vanilla KI method is.

Page 3: “The knockoff framework was first proposed in Barber & Candes (2015) as a data-driven method to control the FDR in variable selection for sparse regression problems.”

Comment: As an outsider, I would really appreciate if the authors say at this point how knockoff works - without a proper understanding on how it works, much of the followup discussion doesn’t make sense.

Page 3: “For two number a and b, let a ∨ b represent min(a, b).”

Comment: I find this notation disturbing. Why?

Page 7: “Specifically, we fine-tune the models using the training samples while ensuring that the non-training samples remain unseen by the models.”

Comment: It is not clear to me why it is necessary to fine-tune the models using the training samples - as they are used in the training already I suppose? Also, such fine-tuning might make the model unnecessarily overfitted on these samples, which makes detecting them easier than what should be the case.

**Questions:**

(1) Why do you choose the L2 norm as the STD score?

(2) Can you provide some statistical guarantee when deciding the number of samples?

---

### Official Review · Reviewer_9C67 · 2024-10-27

**Soundness:** 3
**Presentation:** 3
**Contribution:** 3
**Rating:** 8
**Confidence:** 4

**Summary:**

This work proposes KTD, by casting training data detection (also known as membership inference) as variable selection, allowing them to apply Knockoff Inference (KI) with its strong false discovery rate (FDR) control properties. To improve the performance of this approach, they propose a modification of an established knockoff statistic by computing it using multiple knockoffs instead of just one. They go on to show that under some assumptions this modification leads to an asymptotically optimal (in terms of FDR and power, not efficiency) approach. They demonstrate empirically, that KTD is more effective than vanilla KI and existing training data detection in some settings while crucially not requiring access to a validation set.

**Strengths:**

* New perspective (no validation set, low FDR setting) in an important area.
* KI seems well suited to the problem of training data detection and this work demonstrates how it can be applied (with minimal modifications).
* The paper is well written, assumptions/prerequisites are (mostly) outlined, and the proposed method is easy to understand even without prior knowledge of KI.
* The proposed method seems to scale well with model size and averaging count m. Obtaining strong results in many settings.
* First method to not require access to a validation set including both training and non-training samples (non-training samples are still required)

**Weaknesses:**

* FDR control bounds from Proposition 2 do not seem to hold in general (see Figure 1), and in particular is violated for the important regime of low FDR
* Complete Assumptions for asymptotic optimality (Theorem 1 and 2) are not clearly stated. In particular, Lemma 1 seems to be more of an Assumption than a Lemma (which does not necessarily hold in practice).
* Important experimental setting details are missing, e.g., exact fine-tuning setup.
* Important ablations over the concentration of relevant samples in the training set and the ratio of training to non-training samples in the detection set are missing.
* Related work from membership inference with a similar focus on FDR (Carlini et al., Mireshghallah et al.) and contamination detection (Dekoninck et al. A) is not mentioned or compared.
* Limitations of the work are not discussed, e.g. the need to compute several gradients wrt model weights and thus applicability only to open weight models and the inability to detect training on rephrased data (Dekoninck et al. B).)

**References**
* Carlini et al. "Membership inference attacks from first principles."
* Dekoninck et al. A "ConStat: Performance-Based Contamination Detection in Large Language Models."
* Dekoninck et al.  B"Evading data contamination detection for language models is (too) easy."
* Mireshghallah et al. "Quantifying privacy risks of masked language models using membership inference attacks."

**Questions:**

### Questions
1) Can you discuss why the FDR bound does not generally seem to hold for low q/FDR (See Figure 1) and extend all experiments to the particularly relevant low-FDR regime (1%/0.1%) that would be crucial for the motivating example of legal action.
2) What parameters were used for fine-tuning?  Is any form of “background” data used to dillute the “training” data later checked for? E.g. combining with IFT data such that the checked for training data is only 10%/5%/1% would be a much more realistic setting.
3) How does FDR control change if the portion of training samples compared to non-training samples in the detection set is reduced to 10%/5%/1%?
4) In Figure 2, it seems like the tails of the training samples are quite thick. Can you plot a comparison of just the two tails (positive and negative W_j). Symmetry even there would be required for FDR control at low FDR.
5) Can you extend Figure 4 to focus on the low FDR regime (smaller q) and perhaps a logarithmic x-axis?
### Comments
* Is there a typo in equation 1 and $\vee$ should correspond to the maximum? Otherwise, we would almost always expect FDR and Power >1
* Equation 7 should probably be an argmin (?)
* Definition 2 might be clearer if done elementwise rather than over all variables/training samples jointly, given that this is not leveraged later.
* What does “achieve FDR” mean in Table 2, given that q was set to 1, implying that all “detected” samples can be non-training?
* Can you clarify how training data detection is different from membership inference?

### Conclusion
This work proposes an interesting and novel approach in the context of training data detection/membership inference which works well across a range of settings. However, the crucially important and motivating setting of low FDR, in particular under realistic training and testing settings is not sufficiently investigated. As is, I believe the paper falls slightly short of the acceptance threshold at ICLR, but I would be more than happy to revise my review if my concerns are addressed.

---

> ### Comment · Reviewer_9C67 · 2024-11-22
>
> I want to thank the authors for the substantial effort they put into addressing my questions and comments. While the answers have shown some limitations of the proposed approach, I agree with the authors that they still show its significant promise and highlight the potential for improvements in future work. I have thus raised my score.
>
> I would encourage the authors to include some of the additional experiments in the (Appendix of the) revised paper. In particular, I believe they should highlight that generating "perfect" knockoffs that induce a symmetric knockoff statistic (even in the tails) is challenging in natural language and that Lemma 1 holds only for an ideal knockoff according to Definition 1 and not the (realistic) knockoffs used in the later evaluation. (Including a comment on the (in)feasibility of selecting knockoffs for symmetric statistic distributions in practice).

---

### Official Review · Reviewer_gGP6 · 2024-10-29

**Soundness:** 3
**Presentation:** 4
**Contribution:** 3
**Rating:** 6
**Confidence:** 3

**Summary:**

This paper approaches training data detection as a variable selection problem and introduces a novel Knockoff Inference-based Training Data Detector (KTD). While traditional knockoff inference methods rely on a single knockoff draw, this work extends the approach by incorporating multiple knockoff draws. Through theoretical analysis, the authors demonstrate the asymptotic optimality of their enhanced method.

**Strengths:**

This paper presents a compelling and innovative approach to training data detection. The theoretical foundation is rigorous, demonstrating careful attention to mathematical analysis and proofs. The experimental validation is comprehensive, featuring extensive empirical studies that effectively support the theoretical claims.

**Weaknesses:**

1. Computational Efficiency:
The method requires multiple rounds of rephrasing and inference, resulting in higher computational costs compared to baseline approaches. While these costs are reasonable given the improved performance, the paper would benefit from explicitly discussing this computational trade-off.

2. Performance Limitations:
For small values of q, the False Discovery Rate (FDR) control becomes less strict, which impacts the method's effectiveness. But I think this limitation is acceptable within the broader context of the method's benefits.

3. Novelty Assessment:
The core method is similar to the foundation established by [1,2]. While the paper's primary contribution lies in developing a theoretical framework around this existing idea, the incremental nature of this advancement should be considered when evaluating its impact.


[1] Mattern, Justus, et al. "Membership inference attacks against language models via neighbourhood comparison." arXiv preprint arXiv:2305.18462 (2023).
[2] Fu, Wenjie, et al. "A Probabilistic Fluctuation based Membership Inference Attack for Generative Models." arXiv preprint arXiv:2308.12143 (2023).

**Questions:**

see the weakness

---

### Official Review · Reviewer_a4A6 · 2024-11-05

**Soundness:** 3
**Presentation:** 2
**Contribution:** 2
**Rating:** 6
**Confidence:** 4

**Summary:**

The paper presents a method for detecting if a given set of samples has been used during training of the model, while controlling the false positive rate (e.g., saying a sample is used for training where its actually not). The method (KTD) is phrased as a variable selection problem, enabling the use of existing knock-off inference (KI) techniques. The paper adapts the KI techniques by drawing not a single sample, but multiple samples for each knockoff variable, helping reduce the variance and improve the results. Further, the paper proves that the proposed KTD method preserves the good properties of KI, while showing that as the number of samples approaches infinity, the false positive rate approaches 0 (a desirable property). The method is evaluated on several language models showing promising results.

**Strengths:**

+ The paper addresses an important problem: detecting whether a set of samples has been used for training the model, assuming white-box access to the model (weights, gradient info, etc.).
+ It proposes an adaptation of an existing mechanism (KI), which to my knowledge is the first time this mechanism has been applied to LLMs.
+ The method seems like a clean extension of KI, preserving the good properties and providing optimality at the limit.
+ The method is evaluated on several models.

**Weaknesses:**

- The conceptual contribution is marginal: the KTD method is a somewhat direct adaptation of KI with more than one sample.
- It is unclear what the value of Def 1 is: the requirements cannot be guaranteed on sampling, and it is unclear where it is used in the proofs (I checked the proofs, but perhaps it is in some of the works this paper is citing).
- Not major, but there is something unclear in the problem definition, namely what is D_train. Initially, in Background, X1...Xn are samples to be tested, partitioned into D_train and else. These samples may only be a small part of the training dataset, as it happens in evaluation say. Yet later in 4 (line 181), it defines \theta = Alg(D_train), so D_train is the entire training dataset. Or may be it means that the model is trained on some data and further fine-tuned on D_train. This should be clarified.
- In equation 4, t_j / t is a function used to compute the Ws, yet in equation 7 t is again used, but now it is not a function but its bounding Ws. This should be clarified.
- There is something questionable in the problem formulation: it gives FDR over a set of samples which is determined per equation 7 from a joint bound \tau. Why not solve the problem per single sample one-at-a-time, why does it have to be joint?
- Also given the threshold \tau is jointly computed, it seems possible for one (or two, etc.) difficult (call it adversarial) samples to affect the other (easy) samples which may otherwise require much fewer samples, in some sense requiring much larger overall 'm'. In a sense, the method is not adaptive. This doesn't seem to happen in the evaluation, but I can see a situation where it is possible that one trains/fine-tunes with specially designed-paraphrased adversarial samples which trigger this behavior.
- Experimentally, it seems that the larger the model is, the less one needs the generalization proposed by the paper and the higher values of m as the model memorizes more. For Pythia and GPT-neo (around 1B models), even very few samples seem enough. How would the graph look like with standard 7B models (llama, etc.)? Would m=1 essentially work meaning KI is enough? Can one evaluate a 7B model, there are plenty of those around or perhaps a gemma-2.6b model.

**Questions:**

- My questions related to the paper are listed in the weaknesses above, please address them in the rebuttal.

Other thoughts:

- It may be interesting for further exploration is whether it is possible to fine-tune the model with (adversarialy-paraphrased designed) samples so that the method (KI or KTD) requires many more samples m, without actually reducing accuracy?
- And also, how well your method perform against defenses against membership inference, e.g. Differential Privacy, with certain levels of noise? This is not for this paper, but would be interesting to look at.

---

### Meta-Review · Area_Chair_9v2L · 2024-12-23

**Metareview:**

All reviewers agreed this paper should be accepted: it addresses an important problem, the method has theoretical guarantees on FDR control, and the paper is clearly written. A clear accept. Authors: you've already indicated that you've updated the submission to respond to reviewer changes, if you could double check their comments for any recommendation you may have missed on accident that would be great! The paper will make a great contribution to the conference!

**Additional Comments On Reviewer Discussion:**

All reviewers responded to author feedback, three with short statements thanking the authors and saying they will not change their score. One with suggestions for further experiments and a score increase. All reviewers except Reviewer gGP6 gave extremely detailed reviews.

---

### Decision · Program_Chairs · 2025-01-22

Accept (Poster)